# Comparison of stochastic and deterministic models for *gambiense* sleeping sickness at different spatial scales: A health area analysis in the DRC

Christopher N. Davis[1,2,3]*, Ronald E. Crump[1,3], Samuel A. Sutherland[1,4], Simon E. F. Spencer[1,5], Alice Corbella[1,5], Shampa Chansy[6], Junior Lebuki[6], Erick Mwamba Miaka[6], Kat S. Rock[1,3]

**1** Zeeman Institute for Systems Biology and Infectious Disease Epidemiology Research, The University of Warwick, Coventry, United Kingdom, **2** School of Life Sciences, The University of Warwick, Coventry, United Kingdom, **3** Mathematics Institute, The University of Warwick, Coventry, United Kingdom, **4** Warwick Medical School, The University of Warwick, Coventry, United Kingdom, **5** Department of Statistics, The University of Warwick, Coventry, United Kingdom, **6** Programme National de Lutte contre la Trypanosomiase Humaine Africaine (PNLTHA), Kinshasa, Democratic Republic of the Congo

* C.Davis.7@warwick.ac.uk

**Data Availability Statement:** This study was a re-analysis of the gambiense human African trypanosomiasis (HAT) data that were obtained

## Abstract

The intensification of intervention activities against the fatal vector-borne disease *gambiense* human African trypanosomiasis (gHAT, sleeping sickness) in the last two decades has led to a large decline in the number of annually reported cases. However, while we move closer to achieving the ambitious target of elimination of transmission (EoT) to humans, pockets of infection remain, and it becomes increasingly important to quantitatively assess if different regions are on track for elimination, and where intervention efforts should be focused. We present a previously developed stochastic mathematical model for gHAT in the Democratic Republic of Congo (DRC) and show that this same formulation is able to capture the dynamics of gHAT observed at the health area level (approximately 10,000 people). This analysis was the first time any stochastic gHAT model has been fitted directly to case data and allows us to better quantify the uncertainty in our results. The analysis focuses on utilising a particle filter Markov chain Monte Carlo (MCMC) methodology to fit the model to the data from 16 health areas of Mosango health zone in Kwilu province as a case study. The spatial heterogeneity in cases is reflected in modelling results, where we predict that under the current intervention strategies, the health area of Kinzamba II, which has approximately one third of the health zone's cases, will have the latest expected year for EoT. We find that fitting the analogous deterministic version of the gHAT model using MCMC has substantially faster computation times than fitting the stochastic model using pMCMC, but produces virtually indistinguishable posterior parameterisation. This suggests that expanding health area fitting, to cover more of the DRC, should be done with deterministic fits for efficiency, but with stochastic projections used to capture both the parameter and stochastic variation in case reporting and elimination year estimations.

from the WHO HAT Atlas and are subject to a data sharing agreement. Researchers who meet the criteria for access should apply to the WHO (contact neglected.diseases@who.int or visit https://www.who.int/teams/control-of-neglected-tropical-diseases/human-african-trypanosomiasis/atlas-of-hat). Code and simulated data derived through the re-analysis undertaken in this study are available in Open Science Framework at http://doi.org/10.17605/osf.io/6rfwm.

**Funding:** This work was supported by the Bill and Melinda Gates Foundation (www.gatesfoundation.org) through the Human African Trypanosomiasis Modelling and Economic Predictions for Policy (HAT MEPP) project [INV-005121] (C.N.D., R.E.C., S.A.S, S.E.F.S, and K.S.R.). The funders had no role in study design, data collection and analysis, decision to publish, or preparation of the manuscript.

**Competing interests:** The authors have declared that no competing interests exist.

## Author summary

*Gambiense* human African trypanosomiasis (gHAT, sleeping sickness) is a parasitic infection transmitted by tsetse in sub-Saharan Africa. The distribution of infections is patchy and highly correlated to the regions where humans and tsetse interact. This presents the need for mathematical models trained to the particular regions where cases occur. We show how a stochastic model for gHAT, which captures chance events particularly prominent in small populations or with extremely low infection levels, can be directly calibrated to data from health areas of the Democratic Republic of Congo (DRC) (regions of approximately 10,000 people). This stochastic model fitting approach allows us to understand drivers of transmission in different health areas and subsequently model targeted control interventions within these different health areas. Results for the health areas within the Mosango health zone show that this modelling approach corresponds to results for larger scale modelling, but provides greater detail in the locations where cases occur. By better reflecting the real-world situation in the model, we aim to achieve improved recommendations on how and where to focus efforts and achieve the elimination of gHAT transmission.

## Introduction

Mathematical models have an important role in the ability to quantify progress towards or achievement of location elimination of transmission (EoT) of *gambiense* human African trypanosomiasis (gHAT) [1]. Cases of gHAT—a parasitic infection with a high probability of severe disease and death if left untreated—have greatly declined in the last two decades, predominantly due to ongoing detection and treatment activities in the endemic regions [2, 3]. However, when the target is EoT, it becomes key to estimate the real number of infections, which cannot directly be observed, to understand where transmission is still ongoing and hence where continued intervention efforts are most crucial [4]. It is thus necessary to develop fine-scale models, such that the spatial distribution of infections can be estimated across the endemic regions.

The current spatial distribution of gHAT cases is highly dependent on several geographic factors, such as the density of the disease vector—tsetse (Glossina spp.)—suitability for the parasite, *Trypanosoma brucei gambiense*, and the location of human settlements and places of work [5]. These factors mean the distribution of gHAT endemic regions is highly heterogeneous and confined to geographically distinct regions or "foci", with very few cases detected in areas in between. These foci are typically located in riverine areas of sub-Saharan Africa, where humans and tsetse come into contact [6]. As well as these environmental factors, the local availability of gHAT diagnostics and treatment will also impact the spatial infection distribution. There are long timescales of gHAT infection, which have been estimated at approximately three years before the likely death in the absence of treatment [7]. This means that early diagnosis can substantially reduce the likelihood of onward transmission, by reducing the time people spend infectious and hence the possibility that tsetse will take blood meals on infected people, as well as preventing the worst disease and death in patients. Reductions in the time people spend as infectious, reduce the possibility that tsetse will take blood meals on infected people and cause future transmission [8].

Current gHAT control strategies focus on active and passive screening [9]. Active screening is the use of mobile teams to conduct mass screening of at-risk populations using

serological tests, which detect the presence of antibodies to *Trypanosoma brucei gambiense*. Positive serological tests are followed up with microscopy for parasitological confirmation before treatment is provided. The current treatment options for gHAT are not recommended for the treatment of non-confirmed individuals [10, 11]. Passive screening is where gHAT cases are detected through self-referral to fixed health facilities and relies on training health staff and the availability of diagnostic screening tests to be able to make diagnoses and administer treatment. Additional interventions such as vector control have also been deployed in selected high-prevalence locations to reduce tsetse populations and hence reduce transmission [12], although operational considerations have prevented the use of vector control in many endemic foci, especially in the Democratic Republic of Congo (DRC) which is a large country with most (>57%) of the gHAT case burden for 2021 [2].

The fact that gHAT cases are spatially localised and the interventions to reduce case numbers are implemented in specific locations means that there are great benefits to modelling gHAT at small spatial scales. Several previous studies have only modelled gHAT infections in moderate to large populations. For example, health zones of around 150,000 people in the DRC [8, 13–16], former provinces of the DRC with more than 8,000,000 people [17], foci in Guinea of 14,500 people [18], Chad foci of 40,000 people [19, 20] and foci in Côte d'Ivoire of approximately 300,000 people [21]. These studies fit deterministic models to longitudinal case data, but due to the large populations, there can be large spatial variation within these regions. Some studies have considered modelling individual villages [22, 23], but it is infeasible, both computationally and statistically, to fully fit complex models for all separate endemic villages in a country as large as the DRC.

Here, we consider modelling gHAT infection for DRC health areas (regions of approximately 10,000 people). This is small enough to capture local variation in transmission and to define operationally-appropriate intervention areas, yet not so small as to be too computationally intensive to implement across the endemic region, or mean there is insufficient data for each location to fit a model. The relatively small population sizes coupled with expected gHAT prevalence of <1% meant we chose to use a stochastic model to capture the full range of possible model outcomes and uncertainty since the stochastic effects will be relatively larger compared to larger spatial scale studies [24]. A second advantage of stochastic modelling is that it is straightforward to establish at what time in the simulation that elimination of transmission and elimination of infection is achieved, whereas the analogous deterministic model would require us to specific a proxy threshold for elimination [25]—this is most pertinent to our projections forward in time following model calibration.

In this manuscript, we use the example health zone of Mosango in Kwliu province, DRC, to fit a stochastic model of gHAT infection to its 16 constituent health areas. This methodology updates the work of Crump et al. [8] and combines tools from Spencer [26], with a particle Markov chain Monte Carlo (pMCMC) fitting procedure [27] to allow for calibration of a stochastic transmission model, rather than relying upon fitting a deterministic variant without stochastic, event-driven variation.

We assess the quality of model fitting for health areas and compare the benefits of directly fitting the stochastic model, to using a deterministic model, or the outputs from a stochastic model with posterior parameterisation derived from deterministic fitting. We also compare fitting the model to health zone data with the aggregation of model fitting for constituent health areas. While the fitting methodology is updated, we ensure the process remains robust, automated, and replicable, such that multiple stochastic health area models can be fitted simultaneously. We propose a computationally efficient method to fit health area models for all analysable health areas in the DRC.

## Materials and methods

Ethics Approval was granted by the University of Warwick Biomedical and Scientific Research Ethics Committee (application number BSREC 80/21–22) to use the previously collected DRC country HAT data, provided through the framework of the WHO HAT Atlas [3], in this secondary modelling analysis. No new data collection took place within the scope of this modelling study. Full model code can be found in an Open Science Framework repository at http://doi.org/10.17605/osf.io/6rfwm.

### Data

The World Health Organization (WHO) HAT Atlas data is a record of the globally reported gHAT cases across [2, 3, 28]. For the present study, we have used the data in the HAT Atlas from the DRC, which includes the location of each case, the year of detection, and the mode of detection (either active or passive screening). Where active screening for gHAT has occurred, the total number of people that were screened is given, along with a population estimate for the whole settlement. We used available data for the time period of 2000–2020.

Using the geolocations (latitude and longitude coordinates) of active screening events or passive detections, we assigned the cases to administrative regions. This includes both health zones (typically about 150,000 people) and health areas (sub-units of the health zones, typically about 10,000 people). For this process, we used shape files provided by Nicole Hoff and Cyrus Sinai under a CC-BY licence (current versions can be found at https://data.humdata.org/dataset/drc-health-data) that define the borders between neighbouring health areas. For entries missing geolocations we used the names of villages and health areas to extract the HAT Atlas data, following a similar algorithm to that presented by Crump et al. [8]. Details of the standardised process of translating the HAT Atlas data to our extracted data set of cases by health area, year and screening type for the present study are provided in S1 Text.

In this study, we focus on data for the health zone of Mosango (Fig 1). We chose this health zone as a case study since it is located in the relatively high-burden province of Kwilu, has had annual active screening for gHAT infection and the number of cases detected is geographically diverse, but some cases have been detected in each of the health areas within the health zone. This allows us to compare the results of previously fitting the health zone to the aggregated results of fitting its constituent health areas here. All health areas have sufficient data to perform a model fit (see S1 Text).

### Model

Here we describe the previously developed stochastic model (and its analogous deterministic version) and detail the approach we took to perform fitting of this stochastic model to longitudinal data for the first time.

**Compartmental gHAT model.** We used a previously developed, compartmental, mechanistic model to represent the transmission of the *T. b. gambiense* between humans and tsetse, as transitions of people and tsetse between infection states (see Fig 2 for an illustration of the latest version published by Crump et al. [8]). This model has previously been applied to fitting to data in health zones of the DRC [8], and foci in Chad [20] and Côte d'Ivoire [21]. A susceptible person ($S_{Hi}$) can be bitten by an infected tsetse to become exposed ($E_{Hi}$) before developing stage 1 ($I_{1Hi}$) and subsequently stage 2 ($I_{2Hi}$) gHAT infection, if not detected in screening before progression. Treated and recovering people are represented by $R_{Hi}$. Infected people who are not treated are assumed to die and, in the model, are replaced by a new susceptible person, such that we have a closed population. We also include natural births and deaths in all classes.

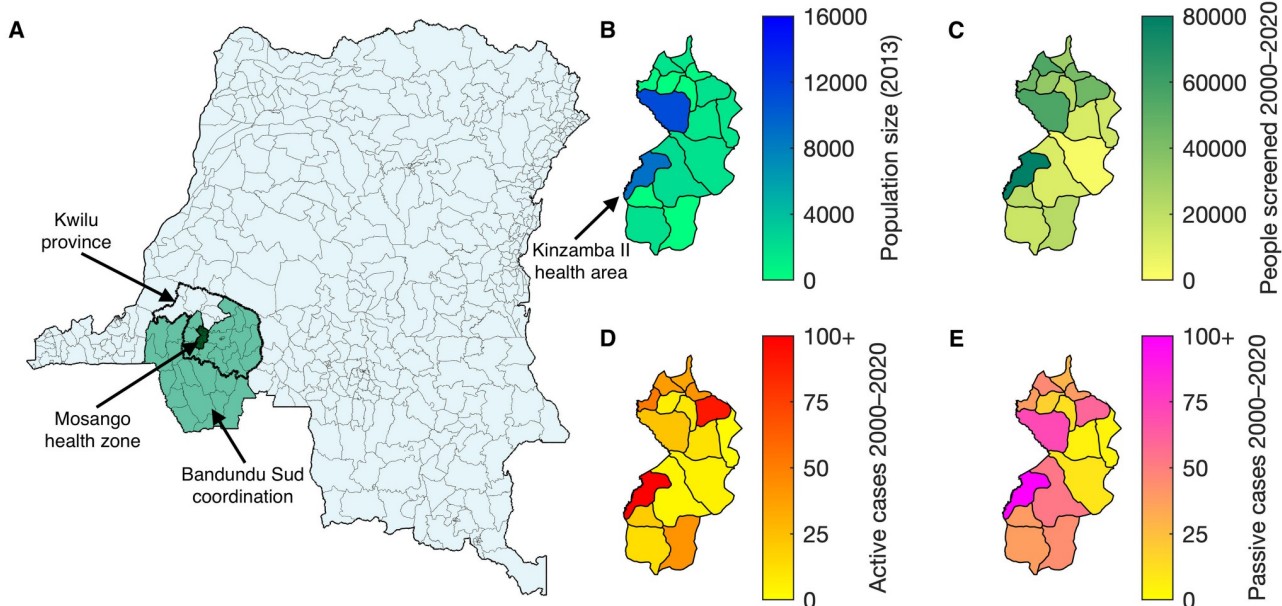

**Fig 1. Data for Mosango health areas.** (A) A map of the location of Mosango, in the Kwilu province and Bandundu Sud coordination of the DRC (coordinations are the large geographic units, similar to provinces, for the organisation of gHAT activities). The map is divided into health area units, where possible, and health zone units otherwise. The Kwilu province is shown by the thick black border, the Bandundu coordination is shown in green and Mosango is dark green. (B–E) The smaller maps to the right are the distribution of the population (data provided by UCLA), the number of people screened (2000–2020) and the number of active and passive cases (2000–2020) across the health areas of Mosango (extracted from the WHO HAT Atlas [3]). The largest number of active and passive cases is 220 and 208 respectively, both in the health area of Kinzamba II. Shapefiles used to produce these maps were provided by Nicole Hoff and Cyrus Sinai under a CC-BY licence (current versions can be found at https://data.humdata.org/dataset/drc-health-data).

Studies have suggested that working-age males commonly work in the riverine areas, which are more densely populated by tsetse, and are therefore more likely to be away from the village working when active screening teams visit [29]. Thus, we include a risk structure in our model. Subscripts $i \in \{1, 4\}$ denote two distinct human risk/behaviour groups in this model, presented as "Model 4" in previous publications [4, 8, 13, 14, 17, 19, 30, 31]. We assume a majority of low-risk people ($i = 1$) that may participate in active screening and a minority of high-risk people ($i = 4$) that both have a larger chance of being infected by tsetse and do not participate in active screening. In this study, we do not consider $i = 2, 3$ however we retain the same notation ($i = 1, 4$) to align with other studies which explore additional risk and behaviour groupings [13, 19, 20].

The proportion of tsetse bites taken on low-risk and high-risk humans ($f_1$ and $f_4$ respectively) depend on both the relative availability and the relative abundance of the two risk groups. High-risk humans are assumed to be $r$ times more likely to receive bites. Therefore, $f_1 = \dfrac{k_1}{k_1 + rk_4}$ and $f_4 = \dfrac{rk_4}{k_1 + rk_4}$. This model structure has been validated in model selection exercises [13, 19, 20].

Tsetse emerge from a pupal phase $P_V$ as susceptible flies $S_V$, but upon taking a blood meal on an infected human, they have a higher probability of being exposed $E_V$ and developing infection $I_V$ if it is the first meal of their lifetime, with previously fed, yet uninfected, tsetse $G_V$ having a lower probability of developing infection when taking an infected blood meal. This phenomenon has been described as the "teneral effect" [32]. For a more realistic extrinsic incubation period, we sub-divide the exposed class into three compartments, $E_{1V}, E_{2V}, E_{3V}$, such

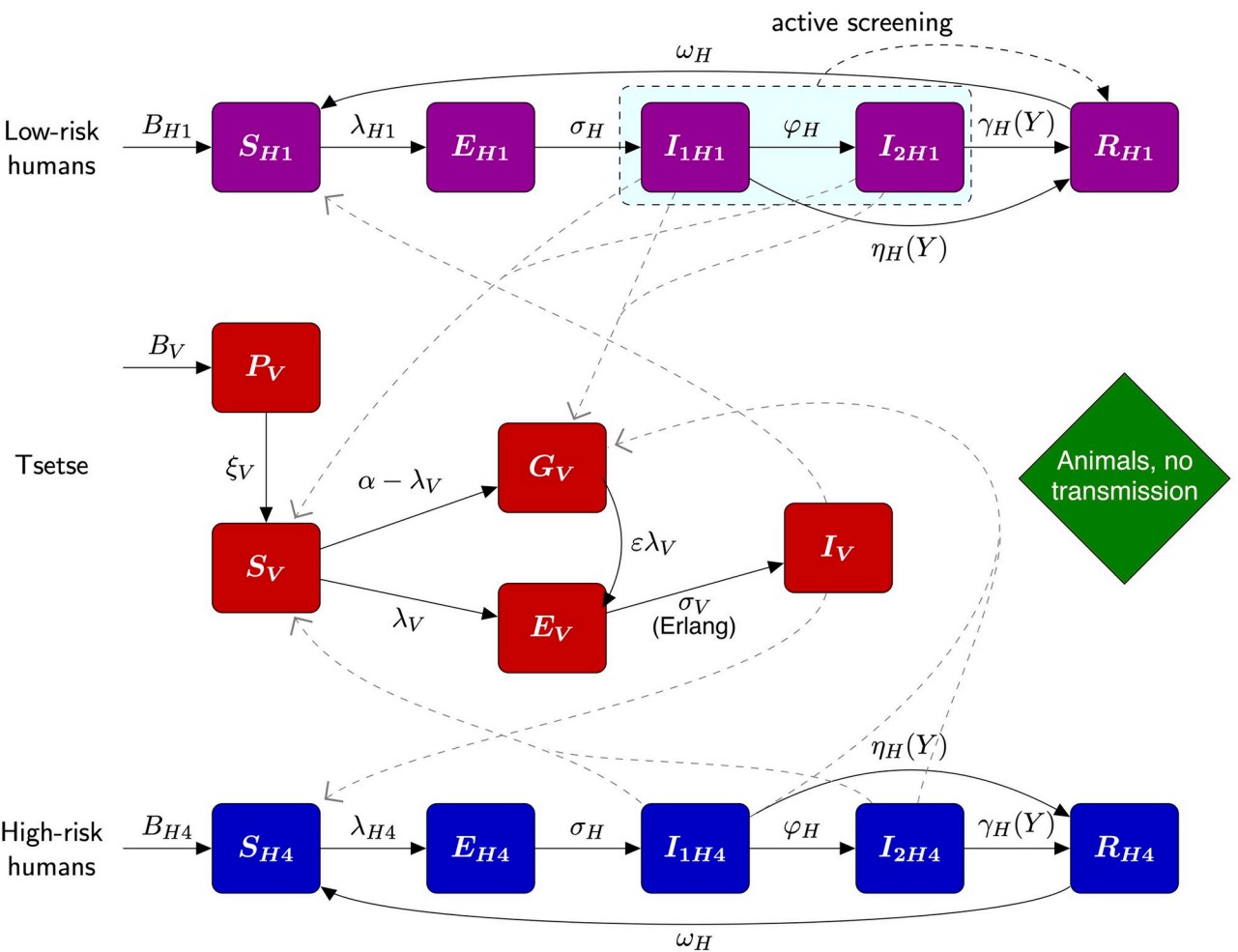

**Fig 2. Compartmental gHAT model.** The purple and blue boxes denote the low-risk and high-risk human infection states, respectively. Low-risk people, who randomly participate in active screening are given the subscript $H1$ and high-risk people, who do not participate, have the subscript $H4$ (this notation is used to align with previously published versions of this model). Tsetse dynamics are given by red boxes. All rates given by arrows are exponentially distributed, except where stated as Erlang. A proportion $f_A = 1 - f_H$ of tsetse bites will be taken on non-human animals, but this does not contribute to the infection dynamics. This figure is adapted from Crump et al. [8] under a CC-BY licence.

that the time a tsetse is exposed is gamma-distributed, rather than exponential. In the present study, we assume that blood meals taken on animals other than humans do not contribute to infections, with animal infection not explicitly modelled. We note that an "animal transmission" variant of our model exists [13, 16, 19, 20], however, there is strong evidence to suggest animals do not form a maintenance reservoir and, if there is some animal transmission in the DRC, the contribution is relatively small and geographically variable—we would expect it to slow progress towards, but not prevent EoT. We therefore only analyse the "no animal transmission" variant here.

We parameterise the model with both fixed values, predominantly biological constants that will not change in different locations, and fitted values for parameters that are location-dependent (Table 1). The fitted parameters are for the basic reproduction number ($R_0$), the proportion of the human population classed as low-risk ($k_1$), the relative number of bites tsetse take on high-risk people ($r$), the diagnostic algorithm specificity in active screening (Spec), the

**Table 1. Model parameterisation.** Notation, a brief description of the parameter, either the fixed value used or the prior distribution used in fitting, and a source for the fixed parameters.

| Notation | Description | Fixed value/Fitted prior * | Source |
|---|---|---|---|
| $N_H$ | Total human population size in 2015 [†] | Fixed for each location | [33] |
| $\mu_H$ | Natural human mortality rate | $5.4795 \times 10^{-5}$ days$^{-1}$ | [34] |
| $B_H$ | Total human birth rate | $= \mu_H N_H$ | |
| $\sigma_H$ | Human incubation rate | 0.0833 days$^{-1}$ | [35] |
| $\varphi_H$ | Stage 1 to 2 progression rate | 0.0019 days$^{-1}$ | [7, 36] |
| $\omega_H$ | Recovery rate or waning-immunity rate | 0.006 days$^{-1}$ | [29] |
| Sens | Active screening diagnostic sensitivity | 0.91 | [37] |
| $B_V$ | Tsetse birth rate (per capita rate of depositing new pupae) [‡] | 0.0505 days$^{-1}$ | [30] |
| $\xi_V$ | Rate of pupal development to adult flies | 0.037 days$^{-1}$ | [30] |
| $K$ | Pupal carrying capacity [§] | $= 111.09 N_H$ | [30] |
| $\mathbb{P}$(pupate) | Probability of a pupa surviving to emerge as an adult fly | 0.75 | [30] |
| $\mu_V$ | Tsetse mortality rate | 0.03 days$^{-1}$ | [35] |
| $\sigma_V$ | Tsetse incubation rate | 0.034 days$^{-1}$ | [38, 39] |
| $\alpha$ | Tsetse bite rate | 0.333 days$^{-1}$ | [11] |
| $p_V$ | Probability of tsetse infection per single infective bite | 0.065 | [35] |
| $\varepsilon$ | Reduced susceptibility factor for non-teneral flies | 0.05 | [13] |
| $f_H$ | Proportion of blood-meals on humans | 0.09 | [40] |
| $R_0$ | Basic reproduction number | $1 + \text{Exp}(10)$ | - |
| $r$ | Relative bites taken on high-risk humans | $1 + \Gamma(3.68, 1.09)$ | - |
| $k_1$ | Proportion of low-risk people | $\text{B}(16.97, 3.23)$ | - |
| $\eta_H^{\text{post}}$ | Treatment rate from stage 1, 1998 onwards | $\Gamma(3.54, 5.32 \times 10^{-5})$ | - |
| $\gamma_H^{\text{post}}$ | Combined treatment and disease-induced death rate from stage 2, 1998 onwards | $\Gamma(2.45, 0.00192)$ | - |
| $b_{\gamma_H^{\text{pre}}}$ | Relative treatment/death rate from stage 2 factor, pre-1998 | $\text{B}(1, 1)$ | - |
| Spec | Active screening diagnostic specificity | $0.998 + (1 - 0.998)\text{B}(7.23, 2.41)$ | - |
| $u(1998)$ | Proportion of stage 2 cases reported from passive screening in 1998 | $\text{B}(20, 40)$ | - |
| $d_{\text{change}}$ | Midpoint year for passive improvement | $2000 + (2017 - 2000)\text{B}(5, 6)$ | - |
| $\eta_{H_{\text{amp}}}$ | Relative improvement in passive stage 1 detection rate | $\Gamma(2.013, 1.049)$ | - |
| $\gamma_{H_{\text{amp}}}$ | Relative improvement in passive stage 2 detection rate | $\Gamma(1.001, 5)$ | - |
| $d_{\text{steep}}$ | Speed of improvement in passive screening detection rate | $\Gamma(39.57, 0.0270)$ | - |
| disp$_{\text{act}}$ | Overdispersion parameter for active detection | $\text{B}(1, 2499)$ | - |
| disp$_{\text{pass}}$ | Overdispersion parameter for passive detection | $\text{B}(1, 35713)$ | - |

*Priors given by Exp(.), $\Gamma$(.) and B(.) are the exponential, gamma (parameterised with shape and scale) and beta distributions, respectively.

[†]The model is internally scaled such that the population size in all years corresponds to the population in 2015 (outputs are back-transformed to reflect an assumed annual population growth rate of 3% across the DRC).

[‡]The value of $B_V$ was chosen to maintain constant population size in the absence of vector control interventions.

[§]The value of $K$ was chosen to reflect the observed bounce back rate.

proportion of stage 2 passive cases reported in 1998 ($u(1998)$), the treatment rate for stage 1 infection from 1998 ($\eta_H^{\text{post}}$), the combined treatment rate and disease-induced death rate from stage 2 infection before 1998 ($\gamma_H^{\text{pre}}$), the combined treatment rate and disease-induced death rate from stage 2 infection from 1998 ($\gamma_H^{\text{post}}$), the post-1998 midpoint year for passive screening improvement ($d_{\text{change}}$), the relative improvement in the passive stage 1 detection rate ($\eta_{H_{\text{amp}}}$), the relative improvement in the passive stage 2 detection rate ($\gamma_{H_{\text{amp}}}$), the speed of improvement in the passive detection rate ($d_{\text{steep}}$), the overdispersion parameter for active detection

(disp$_{act}$), and the overdispersion parameter for passive detection (disp$_{pass}$). All model parameters are given in Table 1.

Several of these parameters ($\eta_H^{post}$, $\gamma_H^{post}$, $\gamma_H^{pre}$, $d_{change}$, $\eta_{Hamp}$, $\gamma_{H_{amp}}$, $d_{steep}$) together combine to parameterise the effect of passive screening (where individuals self-present to fixed health facilities for testing). Before 1998, there was very limited testing capacity with diagnostics not commonly available in health facilities [41], hence we assume a lower detection rate for stage 2 infection before this year and no detection in stage 1, where symptoms are less severe and less specific to gHAT. From 1998, the introduction of the card agglutination test for trypanosomes (CATT) enabled an increased chance of detection for both stage 1 and stage 2 infections [42]. The continued improvement in the availability of diagnostics, including rapid diagnostic tests (RDTs) in the 2010s [43], further improved the passive screening system and therefore the rate of passive detection. Thus, we fully parameterise the rate of leaving stage 1 and stage 2 infections in year $Y$, respectively, as:

$$\eta_H(Y) = \begin{cases} 0, & \text{if } Y < 1998, \\ \eta_H^{post}\left(1 + \frac{\eta_{Hamp}}{1+\exp(-d_{steep}(Y-d_{change}))}\right), & \text{otherwise.} \end{cases} \quad (1)$$

$$\gamma_H(Y) = \begin{cases} \gamma_H^{pre}, & \text{if } Y < 1998, \\ \gamma_H^{post}\left(1 + \frac{\gamma_{Hamp}}{1+\exp(-d_{steep}(Y-d_{change}))}\right), & \text{otherwise.} \end{cases} \quad (2)$$

In addition, active screening, where mobile teams visit endemic villages with the aim of screening as many people as possible for infection, is simulated in the model as an annual event. A random selection of people from the low-risk group $N_{H1}$, using a hypergeometric probability distribution, are tested for gHAT infected, with diagnostic sensitivity and specificity then applied based on binomial probabilities. We assume the diagnostic algorithm has a sensitivity (Sens) of 91% (using a value averaged from Checchi et al. [37]), while the specificity (Spec) has been fitted to match data in the study area. The true positives detected by the model are then assumed to be treated and so moved to the recovering class $R_{H1}$, while false negatives remain undetected. In this study, which focuses on the health zone of Mosango, we additionally assume that post-2015, specificity is 100%. This is because video confirmation of the parasites' presence has been used in addition to other diagnostic tests in this location [44]. Visualisation of the parasite to confirm a case by more than one person means there is little chance of false positives being recorded.

Vector control has been carried out in several areas of the DRC and has been seen to have a large impact on the infection dynamics [12]. Therefore, the model is flexible enough to include deployment of "tiny targets" used to trap tsetse and the robustness of these deterministic fits to data from regions which have implemented vector control during the last 10 years has been previously demonstrated for parts of the DRC, Chad and Côte d'Ivoire [19, 21, 31]. For the health zone of Mosango in the data period 2000–2020, there has been no tsetse control and this is not included in our model fitting presented in the main text, although we do show a fit for the neighbouring health zone, Yasa Bonga—which had vector control starting in 2015, in S2 Text.

We could choose to simulate this model structure with at least two distinct methods: as a deterministic model of ordinary differential equations (ODEs), or as a stochastic model using the tau-leaping algorithm. In this study, we focus on the stochastic model variant (which has previously been used for village-level or health-zone-level simulation [23, 45]), also making comparisons back to the previously-studied deterministic version [8].

The stochastic tau-leaping model is evaluated using a 5-day time step (see S1 Text) and only considers stochastic events in the human component. The tsetse component is updated at each time step using a Runge–Kutta method to implement an ODE system. This choice is due to the difficulty of modelling tsetse explicitly since the exact number of tsetse is unknown in the model; we instead opt to use an effective tsetse density $m_{\text{eff}}$, which is equal to the product of the vector-to-host ratio and the probability of human infection per single infective bite. We non-dimensionalise the vector equations by scaling by $N_H/N_V$, the ratio of human and vector population sizes, such that the effective probability of human infection per single infective tsetse bite ($m_{\text{eff}}$) is defined as $p_H N_V/N_H$, where $p_H$ is the vector-to-human transmission probability. The tsetse ODE simulations will result in a good approximation to a full stochastic simulation since the tsetse dynamics occur on faster timescales to humans and their larger population size means the stochastic effects will be relatively smaller [23].

The full model is described by the events table and ODEs in Table 2.

**Model fitting.** The posteriors of our 14 fitted parameters (see Table 1) are obtained for each location using particle Markov chain Monte Carlo (pMCMC) with the stochastic model described in Table 2. We use the adaptive Metropolis–Hastings random walk algorithm described in Crump et al. [8], but rather than the direct computation of the likelihood, we obtain an estimate using a particle filter (otherwise known as sequential Monte Carlo). We cannot directly compute the likelihood of our stochastic model since we cannot consider the infinite number of possible trajectories the model could take due to the random nature of the model [46].

Particle MCMC is widely used for stochastic model fitting and seeing an increasing use in mathematical epidemiology [47, 48]. Our pMCMC algorithm is the integration of a particle filter to calculate the marginal likelihood into an adaptive Metropolis–Hastings algorithm. Since we cannot consider all stochastic trajectories, we construct an estimate using a set of samples (or "particles"). Using the current parameter set in the MCMC, we simulate the model from the initial state to a data point. At each data point, the particles are assigned weights proportional to the likelihood of the data point. Zero-weight particles are removed and re-distributed according to combine-split resampling [49]. This process prevents particles with no likelihood from propagating forward with no increase in the variance of the marginal likelihood. In addition, if the effective sample size (ESS) of the particle filter falls below the value of half the number of particles, we re-sample the particles according to systematic re-sampling to ensure a high ESS [50]. After the re-sampling, if it occurs, the new set of particles is simulated forward to the next data point and the process is repeated sequentially until the end of the data.

For each particle trajectory (2000–2020), the value of the likelihood calculation is given by Eq 5 below:

$$
\begin{aligned}
LL(\theta|x) &= \log(P(x|\theta)) \\
&\propto \sum_{i=2000}^{2020} \Bigg( \log\bigg[ \text{BetaBin}\bigg( A_{D1}(i) + A_{D2}(i) + A_{DU}(i); z(i), \frac{A_{M1}(i) + A_{M2}(i)}{z(i)}, \text{disp}_{\text{act}} \bigg) \bigg] \\
&\quad + \log\bigg[ \text{Bin}\bigg( A_{D1}(i); A_{D1}(i) + A_{D2}(i), \frac{A_{M1}(i)}{A_{M1}(i) + A_{M2}(i)} \bigg) \bigg] \\
&\quad + \log\bigg[ \text{BetaBin}\bigg( P_{D1}(i) + P_{D2}(i) + P_{DU}(i); N_H, \frac{P_{M1}(i) + P_{M2}(i)}{N_H}, \text{disp}_{\text{pass}} \bigg) \bigg] \\
&\quad + \log\bigg[ \text{Bin}\bigg( P_{D1}(i); P_{D1}(i) + P_{D2}(i), \frac{P_{M1}(i)}{P_{M1}(i) + P_{M2}(i)} \bigg) \bigg] \Bigg),
\end{aligned}
\tag{5}
$$

for model parameterisation $\theta$, and data point $x$. BetaBin($m;n, p, \rho$) gives the beta–binomial

**Table 2. Full model equations.** Events table for the stochastic tau-leaping model for humans and deterministic ODEs for the tsetse dynamics.

Humans:

| Event description | Transition | | Rate |
|---|---|---|---|
| Exposure to infection | $S_{Hi} \to S_{Hi} - 1,$ | $E_{Hi} \to E_{Hi} + 1$ | $\alpha m_{\text{eff}} f_i \dfrac{S_{Hi}}{N_{Hi}} I_V$ |
| Progression to stage 1 | $E_{Hi} \to E_{Hi} - 1,$ | $I_{1Hi} \to I_{1Hi} + 1$ | $\sigma_H E_{Hi}$ |
| Progression to stage 2 | $I_{1Hi} \to I_{1Hi} - 1,$ | $I_{2Hi} \to I_{2Hi} + 1$ | $\varphi_H I_{1Hi}$ |
| Treatment from stage 1 | $I_{1Hi} \to I_{1Hi} - 1,$ | $R_{Hi} \to R_{Hi} + 1$ | $\eta_H(Y) I_{1Hi}$ |
| Treatment or death from stage 2 | $I_{2Hi} \to I_{2Hi} - 1,$ | $R_{Hi} \to R_{Hi} + 1$ | $\gamma_H(Y) I_{2Hi}$ |
| Recovery after treatment | $R_{Hi} \to R_{Hi} - 1,$ | $S_{Hi} \to S_{Hi} + 1$ | $\omega_H R_{Hi}$ |
| Natural birth/death from exposed | $E_{Hi} \to E_{Hi} - 1,$ | $S_{Hi} \to S_{Hi} + 1$ | $\mu_H E_{Hi}$ |
| Natural birth/death from stage 1 | $I_{1Hi} \to 1_{1Hi} - 1,$ | $S_{Hi} \to S_{Hi} + 1$ | $\mu_H I_{1Hi}$ |
| Natural birth/death from stage 2 | $I_{2Hi} \to I_{2Hi} - 1,$ | $S_{Hi} \to S_{Hi} + 1$ | $\mu_H I_{2Hi}$ |
| Natural birth/death from recovering | $R_{Hi} \to R_{Hi} - 1,$ | $S_{Hi} \to S_{Hi} + 1$ | $\mu_H R_{Hi}$ |

(3)

Tsetse:

$$
\begin{aligned}
\frac{dP_V}{dt} &= B_V N_H - \left(\xi_V + \frac{P_V}{K}\right) P_V \\
\frac{dS_V}{dt} &= \xi_V \mathbb{P}(\text{pupate}) P_V - \alpha S_V - \mu_V S_V \\
\frac{dE_{1V}}{dt} &= \alpha(1 - f_T(t)) p_V \left(\sum_i f_i \frac{(I_{1Hi} + I_{2Hi})}{N_{Hi}} + f_A \frac{I_A}{N_A}\right)(S_V + \varepsilon G_V) \\
&\quad - (3\sigma_V + \mu_V + \alpha f_T(t)) E_{1V} \\
\frac{dE_{2V}}{dt} &= 3\sigma_V E_{1V} - (3\sigma_V + \mu_V + \alpha f_T(t)) E_{2V} \\
\frac{dE_{3V}}{dt} &= 3\sigma_V E_{2V} - (3\sigma_V + \mu_V + \alpha f_T(t)) E_{3V} \\
\frac{dI_V}{dt} &= 3\sigma_V E_{3V} - (\mu_V + \alpha f_T(t)) I_V \\
\frac{dG_V}{dt} &= \alpha(1 - f_T(t))\left(1 - p_V \left(\sum_i f_i \frac{(I_{1Hi} + I_{2Hi})}{N_{Hi}} + f_A \frac{I_A}{N_A}\right)\right) S_V \\
&\quad - \alpha\left(f_T(t) + (1 - f_T(t)) p_V \varepsilon \left(\sum_i f_i \frac{(I_{1Hi} + I_{2Hi})}{N_{Hi}} + f_A \frac{I_A}{N_A}\right)\right) G_V \\
&\quad - \mu_V G_V
\end{aligned}
$$

(4)

probability of obtaining $m$ successes out of $n$ trials with probability $p$ and overdispersion parameter $\rho$ and Bin($m;n, p$) gives the binomial probability of obtaining $m$ successes out of $n$ trials with probability $p$. The number of cases detected by passive or active screening in year $i$ is given by $P_{Dj}(i)$ and $A_{Dj}(i)$, where the stage is $j = 1, 2$, or unknown, $U$. Equivalently, $P_{Mj}(i)$ and $A_{Mj}(i)$ are the number of cases detected by passive or active screening in the model.

The probabilities $p$ in the constituent likelihood parts (Eq 5) are: the proportion of the people screened that are detected as active cases, where $z(i)$ is the number of people actively screened in year $i$; the proportion of active cases that are stage 1 disease; the proportion of the total population that are detected as passive cases; and the proportion of passive cases that are stage 1 disease.

We calibrate the number of particles used in the particle filter by comparing the output likelihood for a range of options. We ran 10,000 parameter samples from a previously obtained joint posterior distribution in the particle filter with between 5 and 500 particles (Fig 3A). We thus choose to use 50 particles for our model fitting, since this number is low and so would limit computation time, but it is also high enough to provide stability in the log-likelihood estimation; additional particles showed little change in median value or variance.

As in Crump et al. [8], the MCMC process is fully automated in outputting posteriors and includes three phases run across two independent chains. Firstly, a transient phase with single-site updates only to determine where in parameter space sampling should begin. Secondly, an adaptive phase to start to learn the covariance matrix of the posterior, such that an efficient proposal can be obtained. And thirdly, the sampling phase to obtain the posteriors. The transient phase lasts for 1,000 iterations, the adaptive phase is continued until using the Gelman-Rubin statistic [51], we have $R_{\text{within}}^{(i,j)} \leq 1.1$ and $R_{\text{between}}^{(i)} \leq 1.5$ for parameter $i$ and chain $j$, for a maximum of 100,000 iterations, and the sampling phase ends when $R_{\text{between}}^{(i)} \leq 1.2$ and the ESS of the pMCMC chain is at least 1,000. The ESS is here defined as the minimum value of all the

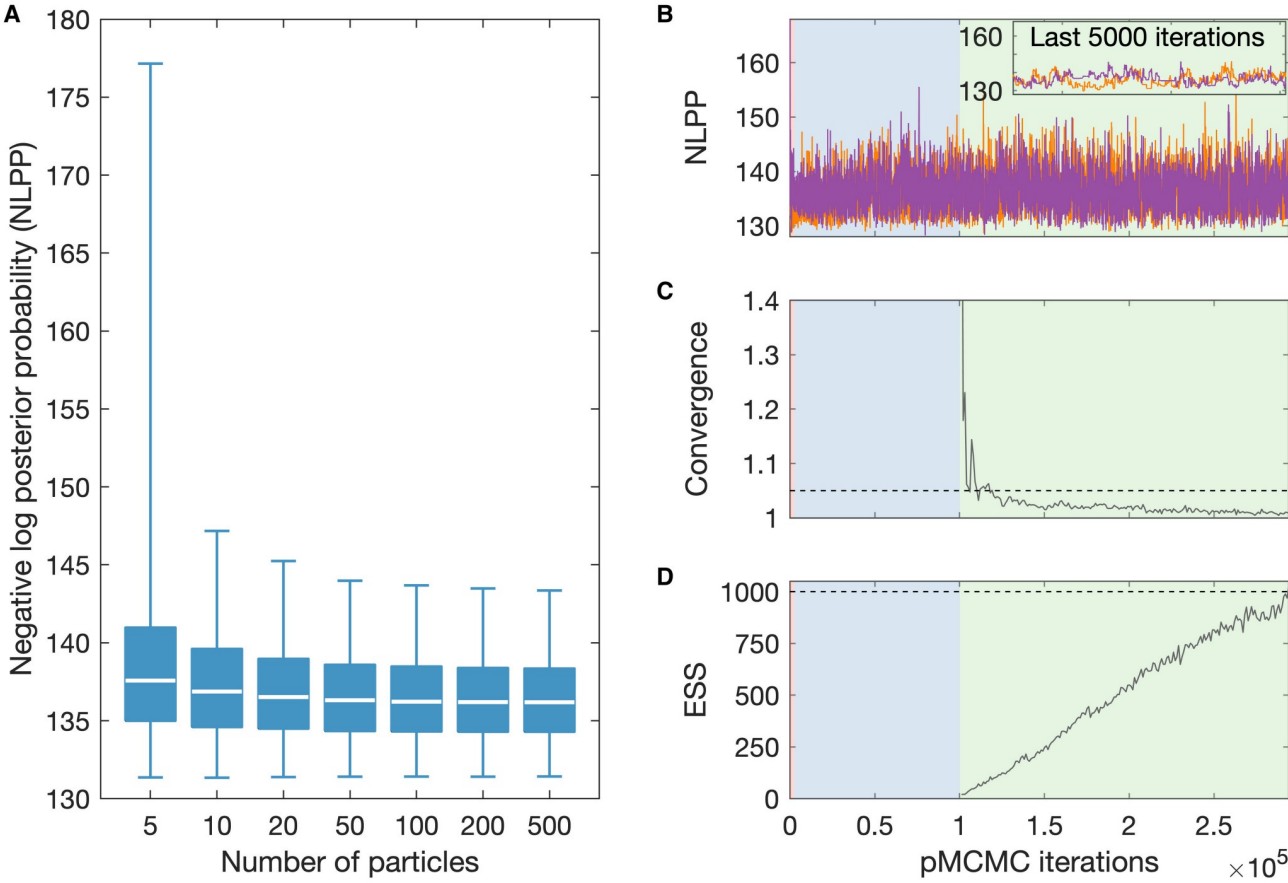

**Fig 3. pMCMC fitting outputs.** (A) Estimates of the negative log posterior probability (NLPP) given by different numbers of particles used by our particle filter. Each box plot consists of 10,000 values consisting of five stochastic realisations from a posterior of 2,000 values. (B) Iterations of example pMCMC chains. Two independent chains and values for the NLPP for 50 particles. (C) The between-chain convergence diagnostic as the number of iterations increases after the burn-in. (D) The effective sample size (ESS) after the burn-in is taken to be the minimum value of the autocorrelation of the thinned sample for all the fitted parameters. Dashed lines on sub-panels C and D show where thresholds for completion lie. Background colours on sub-panels b–D indicate the phase of the pMCMC: red for the (relatively short) transient phase, blue for the adaptive phase, and green for the sampling phase.

fitted parameters, where it is calculated using the autocorrelation of the thinned sample, such that there are 1,000 values for each chain [52].

These phases are shown in an example chain for fitting the health zone of Mosango in Fig 3B. We see that well-mixed chains after the burn-in, throughout the sampling phase, (shown by the green background) result in good convergence below the threshold. The chains terminate when the ESS threshold is reached in a reasonable time.

**Future projections and estimating EoT.** We simulate future projections beyond the end of our data set to 2050, using the parameters obtained in model fitting. We adopt a "continuation" strategy, whereby we maintain the same interventions in the future as the present ones. Annual active screening continues at a coverage equal to the mean annual coverage of the last five years of data and the rate of passive case reporting remains unchanged.

Model outputs include the annual number of active cases given the number of people screened, the number of passive cases detected and the number of new infections. While the actual number of new infections is unknowable (as the timescale from infection to detection can be long and many people are missed in screening), we infer the number of new infections each year both historically and beyond 2020 assuming interventions continue at the same level.

From these simulations, we calculate the probability of EoT (PEoT) as the proportion of model realisations that reach the target of EoT. For calculating PEoT for all the independent health areas of a health zone, we take the product of all health area PEoT values. Hence, we are also able to predict the expected year of EoT, in both health areas and health zones, as the median year EoT is reached in simulations.

## Results

### Fitting to health areas

On performing the fitting process on data for each of the 16 health areas independently, the fitting ended successfully in each case, with an ESS of at least 1,000 and good convergence that satisfied our threshold, such that for all our health area fits, we had $R^{(i)}_{\text{between}} \leq 1.00867$.

Comparing the relevant model outputs back to the data, we have a good correspondence for all health areas. We show example outputs for two of the sixteen health areas of Mosango (Fig 4), chosen as a low-burden health area (Kinzamba I, 15 cases reported in 2000–2020) and the highest-burden health area (Kinzamba II, 428 cases reported in 2000–2020). Outputs for all additional health areas are shown in S2 Text.

Fitting the stochastic model took three to ten times as long to meet our fitting cessation criteria compared to fitting the deterministic model to the same data, with both computations performed on the same cluster with the deterministic code parallelised over two cores and the stochastic code over ten cores. The longest computation times occurred when substantially more MCMC iterations were required to meet the ESS threshold, and the shorter times were when no additional sampling was required beyond the number needed for deterministic fitting (typically <200, 000 iterations).

For both example health areas, the expected trend in reported cases is well captured with almost all data points falling within the displayed 95% credible intervals. Generally, higher historical screening coverage resulted in higher active case reporting, while passive case numbers are more stable, but with a decline throughout the study period, as the underlying expected number of new infections declines. The expected number of new infections in the model also decreases with these trends.

The basic reproduction number $R_0$ is a fitted parameter of the model, which defines the expected number of new infections generated from a single infection in a susceptible population. For gHAT, this number is very close to, but greater than one [8, 13, 53]. This can be

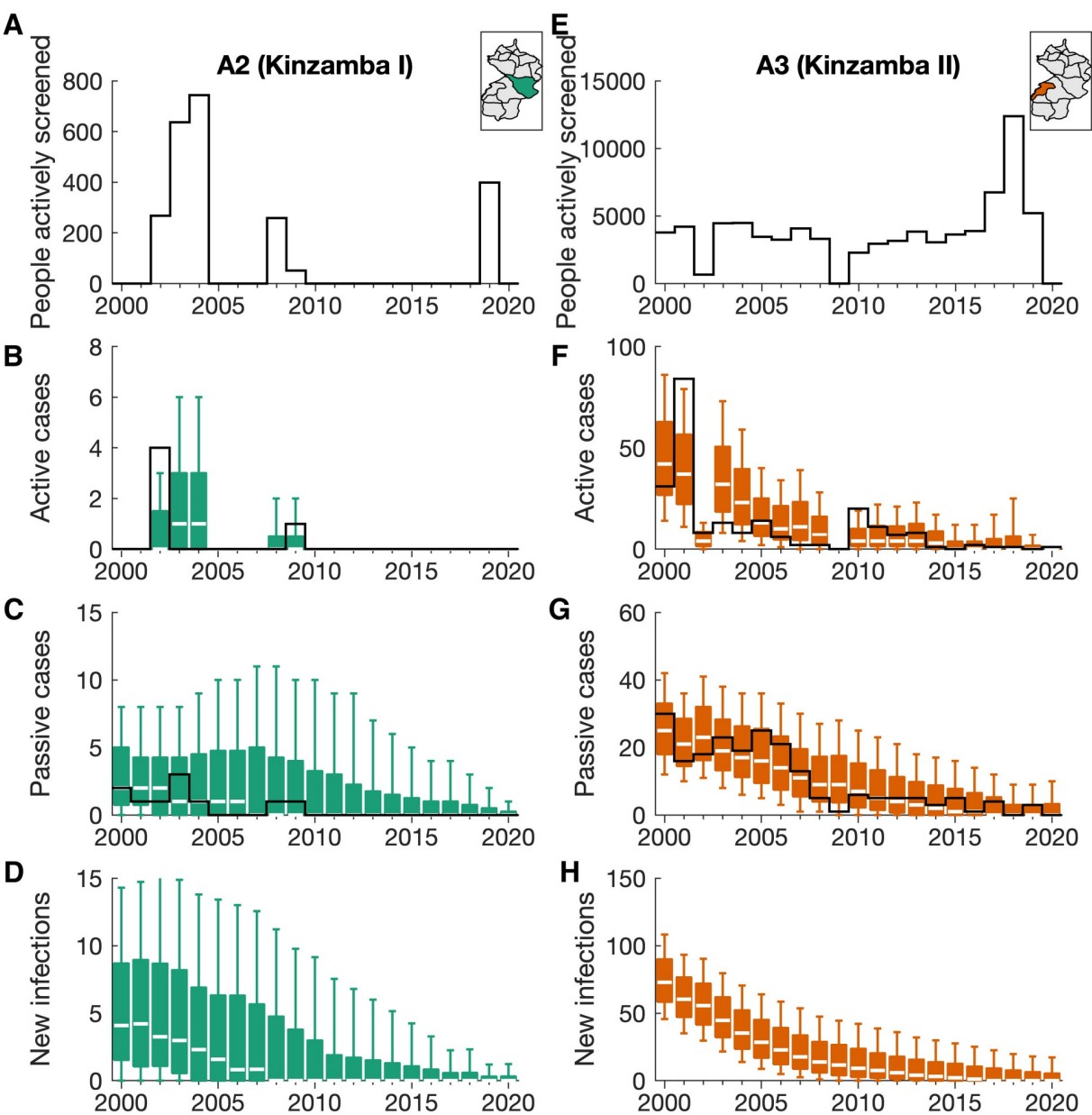

**Fig 4. Model fitting outputs in example health areas.** (A and E) Data for the number of people actively screened and the active and passive cases reported are displayed as solid black lines for two health areas: Kinzamba I and Kinzamba II (labelled as A2 and A3 respectively in this analysis). (B–D and F–H) The modelling outputs for active and passive cases and the number of new infections in each year are given as coloured box plots. The median value of the box plots is shown as a white line, with outer boxes and whiskers representing 50% and 95% credible intervals respectively. No data line is shown for new infections, since this cannot be directly observed. Maps of the health area locations with the Mosango health zone are shown in the top right of each column. Shapefiles used to produce these maps were provided by Nicole Hoff and Cyrus Sinai under a CC-BY licence (current versions can be found at https://data.humdata.org/dataset/drc-health-data).

explained by the fact that gHAT is a slow-moving endemic disease and so infection only needs to cause one more (over several years) to maintain endemic levels of transmission. Several factors will affect the local basic reproduction number for each health area, such as the number of tsetse in the location or the geography of the region defining the contact patterns and hence transmission rate of infection [3].

The geographical distribution of $R_0$ in Mosango health zone shows there is substantial variation between health areas (Fig 5), hence justifying the use of a finer-scale model. Kinzamba II health area (A3) has the largest median value with $R_0 = 1.04$ and the $R_0$ posterior distribution is shifted to be larger than most other health areas. This corresponds with the largest number of cases being detected in this health area. Health areas with the fewest cases, such as Kasay (A1), have the lowest $R_0$ values. However, the median $R_0$ values from the fitting show all health areas have $1 \leq R_0 \leq 1.04$. In all cases, the posterior has greatly deviated from the exponential (but relatively flat) prior distribution.

Values for the parameter $u(1998)$, which is the proportion of stage 2 cases reported from passive screening in 1998, are much more similar across the health zone. The median values are in the range $0.29 \leq u(1998) \leq 0.37$ and the posterior distributions all substantially overlap. This parameter determines the level of under-reporting of infection since it defines the proportion of late-stage cases detected in passive screening, of those that are not detected by active screening. Due to similar access to fixed health facilities across the health zone, it is logical that there would not be a major variation in this parameter across the health zone. For most health areas there is no significant deviation from our informed prior distributions, indicating there

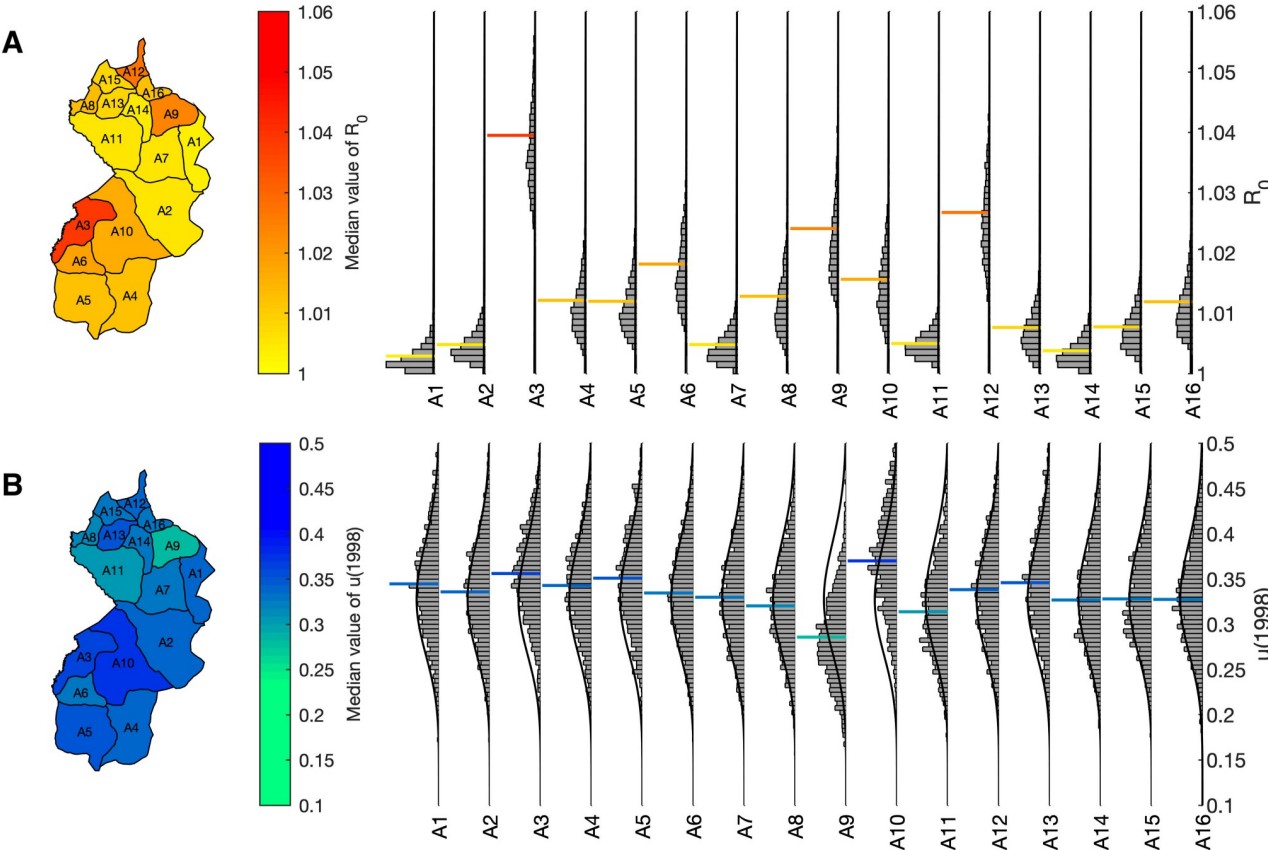

**Fig 5. Geographical distribution of model parameters in Mosango health zone.** The geographical distribution of the median values for: (A) the basic reproduction number $R_0$ and (B) the proportion of stage 2 cases reported from passive screening $u(1998)$ is shown in maps of the health zone. Each health area is labelled A1–A16, with the full posterior distribution given in histograms on the right. The median value is shown by a horizontal coloured line, aligned with the color scale of the map. The prior distribution is shown on the histograms as a solid black line, although with a relatively low probability density in the displayed region for the $R_0$ histograms. It is the same for each health area. The names of health areas A1–A16 are given in S1 Text. Shapefiles used to produce this map were provided by Nicole Hoff and Cyrus Sinai under a CC-BY licence (current versions can be found at https://data.humdata.org/dataset/drc-health-data).

may be insufficient data to show difference from these distributions. However, there are examples where model fitting highlights spatial differences. The reduced value of the median of the posterior distribution in health area A9 (Kumbi Mbwana) indicates passive screening is detecting fewer than expected cases here, given the relatively large $R_0$ value, whereas the larger posterior values in A10 (Mangungu) indicate passive screening is performing well. These results contrast with the health area with most data, A3 (Kinzamba II), where the median estimated value of $u(1998)$ is only slightly higher than the prior.

## Comparison of fitting approaches

In addition to fitting the stochastic model to the health area data, we performed that same fitting methodology with data from the full health zone, and also fit the deterministic model variant to the health zone data using the original methodology of Crump et al. [8]. To compare between the three approaches, we aggregated together the health area data by taking the sum of the output model realisations for each health area, to create health zone distributions.

All three fitting approaches provide very similar outputs that capture the dynamics seen in the data, providing confidence that all of these methods are appropriate for model fitting (Fig 6). In particular, the health zone fitting shows there is very little difference in the median between outputs of the deterministic and stochastic model variants, shown as blue and red boxes respectively in Fig 6. The stochastic model variant does indicate more uncertainty in the number of new infections since the random nature of the model demonstrates a wider range of outcomes can still match the data. There is also more uncertainty in the stochastic model in later years where gHAT is closer to elimination, shown by the longer period for the probability of EoT to increase from zero to one. Hence, this provides evidence that using the stochastic model is more appropriate for low infection numbers, as the uncertainty is representative. This is particularly true when considering the smaller population sizes of single health areas (Fig 7). The larger uncertainty in the versions that use stochastic projections is important for projecting cases in future years, where we would expect a further decline, given current intervention strategies, and more realistically captures the uncertainty in when EoT could occur. We also note that using the posterior of the deterministic fitting in the stochastic model projections provides outputs more similar to the full stochastic output, rather than the deterministic one (Fig 7, see S2 Text for aggregated health area results). This indicates that there are greater fundamental differences between the deterministic and stochastic model variants than can be accounted for in model fitting.

For modelling at different spatial scales, while the outputs are very similar for the health zone and aggregated health areas, it is clear that there is an additional benefit of using a more fine-scaled approach (Fig 6). The aggregation of health area models provides the best fit to the health zone data (green boxes in Fig 6). Some information is lost on the location of active screenings and cases when the data is combined for the purpose of model fitting to health zones, so the more nuanced health area fitting process is more likely to recover the infection dynamics.

## Future projections and estimating EoT

We also compare differences in health area and health zone fitting, by considering projections for future infections and specifically the probability of EoT. For each health area, we calculate the annual probability of EoT (PEoT) as the proportion of samples where EoT has been achieved, assuming continued interventions in future years. Since EoT is defined as when there are no future transmission events, this requires simulating the model beyond just the

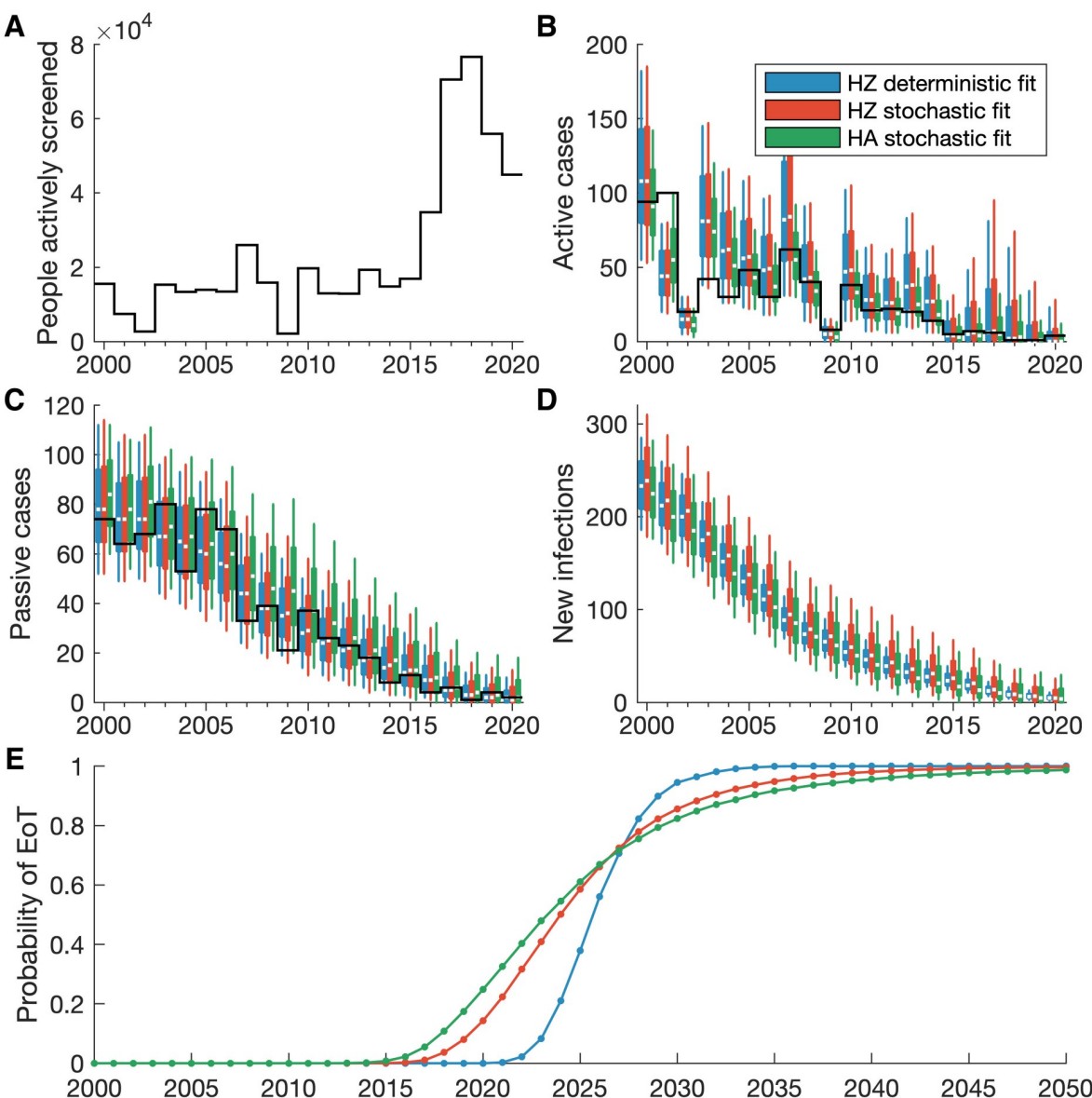

**Fig 6. Model fitting outputs for Mosango health zone.** (A–D) Data for (A) the number of people actively screened and (B) the active and (C) passive cases reported is displayed as solid black lines. Since new infections are not observed there is no data in panel D. The coloured box plots show annual model outputs for different fitting approaches, where blue boxes show the deterministic health zone model, red shows the stochastic health zone model, and green shows the aggregated stochastic health area models. The median value of the box plots is shown as a white line, with box plot whiskers representing 95% credible intervals. (E) The bottom panel shows the probability of EoT from 2000–2050 assuming a continuation of interventions.

first year there are no new infections to ensure people infected for long periods do not cause onward transmission much later.

We see that in each of the health areas of Mosango, there is a very high PEoT by 2030 (Fig 8A). Each health area in isolation has a PEoT by 2030 greater than 0.94, with the lowest in Kinzamba II (health area A3). However, the probability of achieving EoT in all the health areas, and thus the entire health zone, will be substantially lower as it is equal to the product of the PEoTs. Indeed the PEoT in all health areas by 2030 is 0.82 (solid black line in Fig 8A), which is

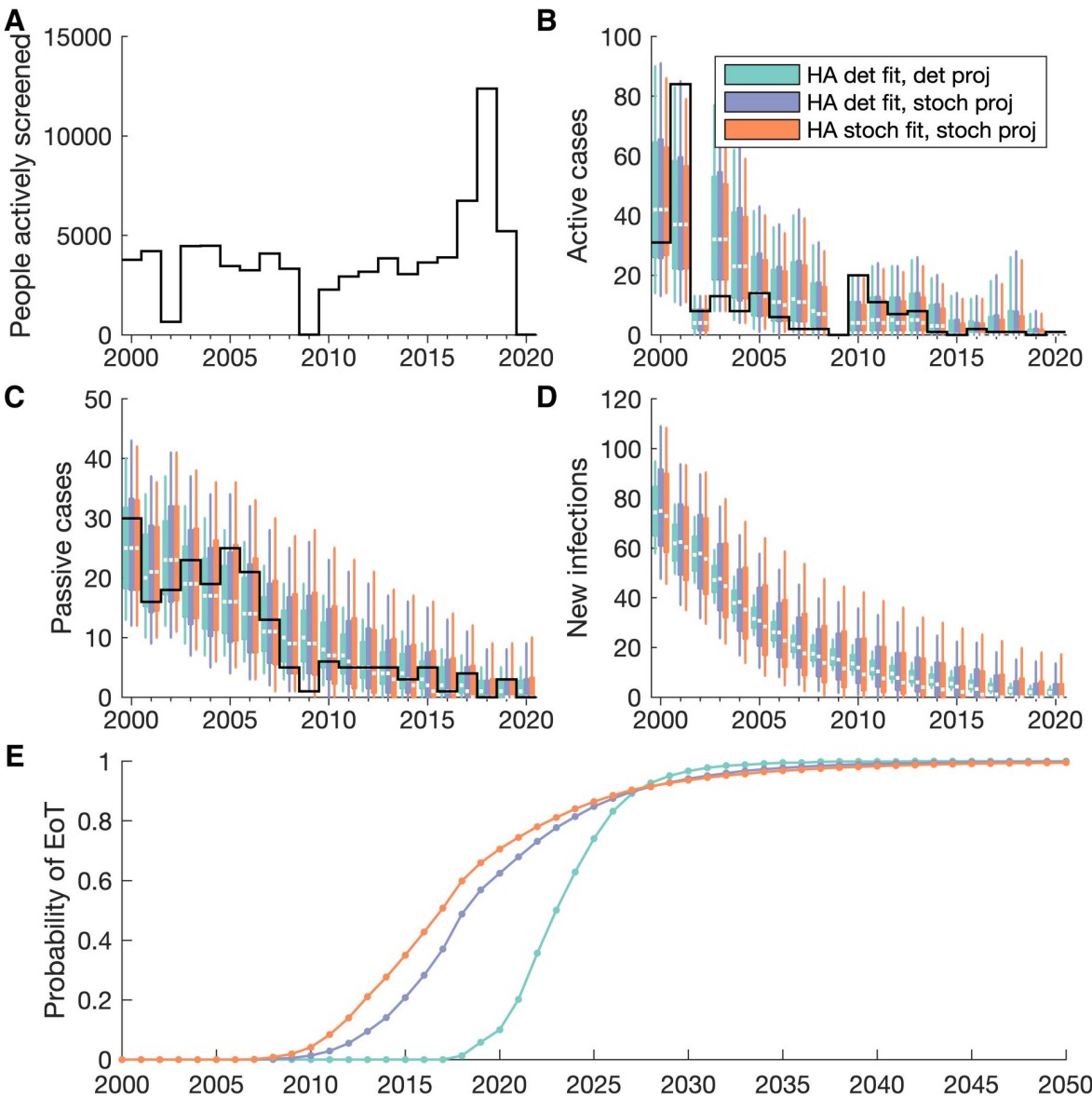

**Fig 7. Model fitting outputs for Kinzamba II (A3) health area.** (A–D) Data for the number of people actively screened and the active and passive cases reported is shown by solid black lines. Since new infections are not observed there is no data in panel D. Coloured box plots show annual model outputs for different methods for both model fitting and model projections. The light blue boxes show the deterministic model fitting with deterministic projections, the light purple shows the deterministic model fitting with stochastic projections, and the light red shows the stochastic model fitting with stochastic projections. The median value of the box plots is shown as a white line, with box plot whiskers representing 95% credible intervals. (E) The bottom panel shows the probability of EoT from 2000–2050 assuming a continuation of interventions.

comparable to the PEoT calculated for the health zone model, which is 0.86 for 2030 (dotted grey line in Fig 8A).

We also display the expected year of EoT, calculated as the median year from all model realisations (Fig 8B–8D). For each health area, the expected years of EoT are all in the past. This is consistent with the fact that when considering a smaller area there is a higher chance of elimination in that area only. Furthermore, while EoT may have occurred in health areas (i.e. the

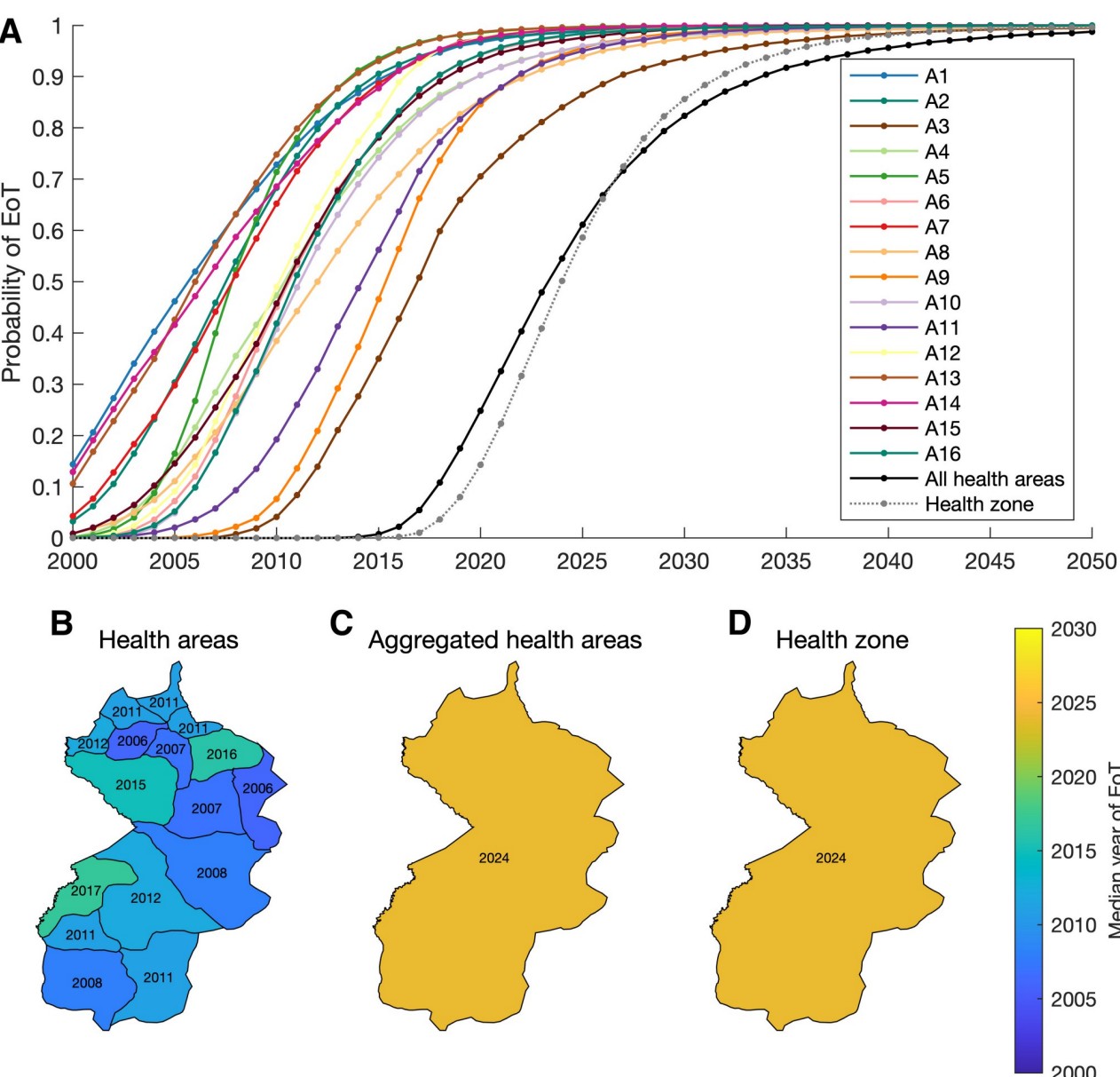

**Fig 8. Elimination of transmission (EoT) in Mosango.** (A) Probabilities of EoT for health area by a given year are shown by coloured lines in the top panel, with aggregated health area results shown as a black line and separate health zone model results shown by a dotted grey line. (B–D) The below panel shows maps of Mosango with median expected years of EoT given by colour. The three maps show three approaches for displaying EoT: (B) for each individual health area, (C) for the health zone using the health area models, and (D) for the health zone using the health zone model. Shapefiles used to produce this map were provided by Nicole Hoff and Cyrus Sinai under a CC-BY licence (current versions can be found at https://data.humdata. org/dataset/drc-health-data).

last transmission events may have already occurred), there could still be undetected infections, that are yet to be reported as cases due to the long infection time scales of gHAT [4]. Indeed, while local EoT may have already occurred in selected health areas, we still predict a later median year of 2024 for EoT for all health areas of Mosango, which aligns with the results for modelling at the health zone level. We also re-iterate that this is to have a 50% probability of EoT and to have higher confidence that EoT has truly occurred, it would be recommended for

vertical interventions to remain in place for several more years, as would also be recommended based on current WHO guidelines for active screening.

## Discussion

By implementing a pMCMC fitting algorithm, we have been able to fit a stochastic infection model to historical gHAT screening and case data. This process is more complex and computationally intensive than fitting a deterministic model with a more traditional MCMC approach. With the particle filter, we can account for stochastic variations in the infection dynamics with improved realism. Moreover, pMCMC allows us to fit the multiple parameters of our stochastic model in a reasonable time frame, rather than relying on the deterministic analogue.

Previous modelling work has used the parameters determined from fitting a deterministic model to make projections of future infection dynamics in a stochastic model variant [4, 15, 45]. We have shown here that this gives a good approximation to using the parameters obtained from stochastic fitting. In particular, we show that in fitting different models to health zone data for Mosango, both deterministic and stochastic models produce very similar results (Fig 6). However we note that results can diverge more in the future projections in the approach to elimination (see S2 Text).

In fitting stochastic models suited to where there are low number of infections, we have also expanded the range of previously modelled locations, to include the small-scale approach of health area modelling. This new modelling at a health area spatial scale provides highly specific results that capture the full heterogeneity and focal nature of gHAT infections. We show that by using the information in the data on the health area location for screening activities and cases, we can obtain a better fit to the data (Fig 6), since screening activities are directly mapped to the specific health areas they occurred in.

The main benefit of health area modelling, however, will be in future projections. A more fine-scaled approach will bring added realism into modelling by more accurately reflecting proposed scenarios and providing more flexibility in the scenarios that can be modelled. For example, increasing active screening coverage in a specific region of a health zone, or only implementing vector control in the relevant high-tsetse density areas can now be modelled, as opposed to blanket changes across the whole health zone (see S2 Text).

We choose to model all health areas independently since we know that gHAT transmission is very localised and in many instances there will only be minimal movement of both people and tsetse between neighbouring health areas [5]. However, this is not uniformly the case, and there could be importations of infection between neighbouring health areas, which are not considered. Future work would consider correlations between infections in health areas that share a border along a river, a typical habitat for tsetse [6], since here the same tsetse population would be infecting humans from both health areas.

In developing our model, we also choose not to consider the presence of possible animal reservoirs or transmission from asymptomatic humans. While *T.b. gambiense* can be found in animals [54], evidence is inconclusive to this contribution to the transmission cycle with humans [55]. Previous modelling has shown that there is some evidence that animals may contribute to transmission in some areas, however, it is extremely unlikely that animals could sustain transmission without humans [16, 20]. Modelling has not yet been able to evaluate the evidence for self-curing asymptomatic human infections on transmission and elimination through matching to data, however, an asymptomatic model sensitivity analysis has been performed [56]. The modelling and fitting methods presented here could be modified readily to include the animal transmission or asymptomatic model variations, however, this would

further increase the time to run the simulation, as an additional two or five model parameters would need to be estimated through the pMCMC for the two models respectively.

Future work should use the principles demonstrated here and expand to provide fitted models for all health areas in the DRC that have sufficient data to be confident in the results. One drawback of this approach will be the lack of data across all endemic regions, particularly as we approach elimination. In Mosango the minimum number of data points we have was 7 (in health area A1, Kasay), however, it is highly likely that some health areas may have fewer data points than this and could render fitting unreliable in those locations. In this instance, options that could overcome this issue include an amalgamation of neighbouring health areas of similar (believed) prevalence to create sub-health-zone but larger-than-health-area sized regions for an analysis—this approach is demonstrated for the two health areas with the fewest data points in Mosango in S1 Text. For some locations, it may not be recommendable to attempt sub-health-zone level fitting due to data limitations.

Our model uses data points on active and passive cases and the infection state of those cases detected (stage 1 or stage 2), where available, to determine the likelihood and thus inform the model parameters. Since 2020, the DRC have expanded first-line treatments for gHAT patients to include fexinidazole [10, 57], it is no longer necessary to determine the infection stage since the treatment is suitable for both stages and the process of determining infection stage, visualising parasites in cerebrospinal fluid, requires a lumbar puncture and is unpleasant for the patients. While this is a great benefit for patients, and results in a reduced incentive to avoid treatment, there is a loss of information in people's infection status. For the purpose of model fitting, more information is always beneficial to better match reality [17] and may lead to deterioration in our ability to fit models unless additional information is recorded such as infection symptoms, as to give a proxy for this lost stage information.

Additionally, we note that fitting models for all health areas is more computationally intensive than fitting to health zones. This is in part due to the fact that there are many more health areas of the DRC and so there are simply more locations; approximately 1200 analysable health areas based on the 2000–2020 HAT Atlas data, assuming a minimum of 10 data points required for fitting, currently available shape files to define health area boundaries, and good estimates of health area population sizes. Also, the stochastic fitting methodology is more intensive than deterministic fitting if we choose to fit with this methodology. In the pMCMC, we have to simulate multiple trajectories to be able to evaluate the likelihood of a parameter set rather than the single deterministic trajectory. The model fitting procedure presented here is fully automated and parallelised to optimise performance on a computer cluster, so while compute times may be longer (three to ten times as long, using five times the number of cores), there is little additional work for the user. However, we have demonstrated in this manuscript that, for gHAT, fitting using the deterministic model variant provides a close approximation (Fig 7), which would negate the necessity for this additional computation time. We note that this will not be true for disease modelling generally; the long time scales of gHAT infection, slow disease progression and small critical community size are likely to make this approximation valid [23].

This study has focused on the computational methodology to allow the gHAT transmission model fitting to health area data in the DRC and does not seek here to make any specific policy recommendations. Our projections simulated from 2020 onwards assume a status quo for interventions and we do not seek to assess the possible impact of changes to interventions. Our study does lay out an appropriate framework with which to evaluate future health-area-level strategy in further analyses, ensuring models are suitably calibrated to local data and that they can provide very granular recommendations, extending the health zone projection and health economic evaluations produced previously [31, 58].

## Conclusion

In demonstrating that a mechanistic, stochastic infection model can be fitted to different health area data in the DRC, we have added a new framework that can provide highly detailed calibration and projections of future gHAT infections for a range of intervention scenarios. The similarity in fitting stochastic models to data and simulating stochastic models, but using deterministic variants for fitting, validates the approach of previous studies and indicates that stochastic fitting may not always be necessary. The importance of stochastic models however remains clear, in determining the time to elimination of transmission and our uncertainty in this.

The greater nuance afforded by smaller-scale health area modelling will further refine the ability to accurately forecast future infections for intervention scenarios, allowing policy-makers to better understand the cost and impact of proposed interventions.

## Supporting information

**S1 Text. Additional methods.** An expansion of the mathematical modelling methodology. (PDF)

**S2 Text. Additional model outputs.** Further comparisons of modelling approaches and full posteriors for health area fitting. (PDF)

## Acknowledgments

The authors thank PNLTHA-DRC for original data collection and WHO for data access (in the framework of the WHO HAT Atlas [3]). We thank Cyrus Sinai and Nicole Hoff from UCLA Fielding School of Public Health for providing health zone-level shape files for the DRC and health area-level for the former Bandundu province (current versions can be found at https://data.humdata.org/dataset/drc-health-data). We also thank the HAT MEPP scientific project manager Emily Crowley for her support in project organisation and proof-reading this manuscript.

## Rights retention statement

For the purpose of open access, the author has applied a Creative Commons Attribution (CC-BY) licence to any Author Accepted Manuscript version arising from this submission.

## Author Contributions

**Conceptualization:** Kat S. Rock.

**Data curation:** Shampa Chansy, Junior Lebuki, Erick Mwamba Miaka.

**Formal analysis:** Christopher N. Davis.

**Funding acquisition:** Kat S. Rock.

**Investigation:** Christopher N. Davis.

**Methodology:** Christopher N. Davis, Ronald E. Crump, Samuel A. Sutherland, Simon E. F. Spencer, Alice Corbella, Kat S. Rock.

**Project administration:** Christopher N. Davis, Kat S. Rock.

**Supervision:** Kat S. Rock.

**Visualization:** Christopher N. Davis.

**Writing – original draft:** Christopher N. Davis.

**Writing – review & editing:** Christopher N. Davis, Ronald E. Crump, Samuel A. Sutherland, Simon E. F. Spencer, Alice Corbella, Shampa Chansy, Junior Lebuki, Erick Mwamba Miaka, Kat S. Rock.

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
