## [Decision Letter · Decision Letter 0]

21 Dec 2023

Dear Dr. Davis,

Thank you very much for submitting your manuscript "Comparison of stochastic and deterministic models for *gambiense* sleeping sickness at different spatial scales: A health area analysis in the DRC" for consideration at PLOS Computational Biology. As with all papers reviewed by the journal, your manuscript was reviewed by members of the editorial board and by several independent reviewers. The reviewers appreciated the attention to an important topic. Based on the reviews, we are likely to accept this manuscript for publication, providing that you modify the manuscript according to the review recommendations.

Sincerely,

Alex Perkins

Academic Editor

PLOS Computational Biology

Thomas Leitner

Section Editor

PLOS Computational Biology

Reviewer's Responses to Questions

**Comments to the Authors:**

Reviewer #1: Summary of the work:

This study compares deterministic and stochastic calibration methods for Gambian human African trypanosomiasis. When dealing with large populations, the research finds that deterministic model calibration closely mirrors the results obtained from stochastic models. This alignment is attributed to fundamental disease dynamics characteristics in gHAT, such as extended infection time scales, slow disease progression, and a small critical community size. This insight offers a practical strategy for reducing computational costs: initiate calibration in a larger population using deterministic models and then apply the obtained parameters to a stochastic model to assess long-term stochastic effects on smaller populations.

I have both minor and major comments as follows:

Minor comments:

A scheme should be though out for identifying sub-figures (use A, B, C, D for example). Right now it is not obvious by reading the caption which sub-figure is being referred to.

In the methods section, give the equation between lines 246 and 247 a number. It seems like that equation number is out of the page. The second comment is regarding the description of terms in this equation. Define Bin(m;n,p) as well along the same footing as BetBin(m;n,p,rho). Can you also explain the reason for each denominator in the likelihood equation in the text just as you explain z(i)?

In the discussion section, line number 426, the word using comes twice at the beginning of this line.

Line 505 . the word ‘no’ should be ‘not’.

In general please run a spell check and also look for repeating words etc.

Major comment:

It would be beneficial if the authors could demonstrate the robustness of their findings by varying disease parameters. The use of toy data to examine the point at which calibration on deterministic and stochastic models diverge in small community sizes could help strengthen their argument.

Reviewer #2: Thanks to the editor and authors for the chance to review this interesting manuscript. The authors present a stochastic version of a previously published deterministic model of gHAT, and use PMCMC to fit this model to smaller administrative regions than implemented previously. The use of the stochastic model is important here since gHAT transmission can be sustained at very low prevalence, so stochastic effects can become important in disease control. Overall I thought that the manuscript was well-structured and convincing in its central arguments, and that the model presented here will help advance our understanding of gHAT dynamics.

I think that there are two main things the authors could do to increase the impact of this work:

1. include a fit of the model to a health zone other than the primary one used in model development. This would demonstrate the flexibility of the model, and comparison of the fitting process in two different contexts could lead to additional insights into how to best use this model. Implementation in a zone where tsetse control was used could be particularly illuminating.

2. the Open Science repository of code should be mentioned in the main text of the paper, and the information and structure of the repository should be improved to facilitate the use of this code by other researchers. Some ideas for how to do this are available from these workshop notes: https://epiverse-trace.github.io/research-compendium/beforestart.html

As a smaller issue, I checked some of the sources in table 1 and found it hard to figure out the justification for some of the parameters. For instance, the active screening sensitivity has a citation to a paper that calculates sensitivity for specific regions not including DRC. This seems to contradict line 191 saying that sensitivity is “fitted to match data in the study area”. The authors seem to be using the estimate for the Republic of Congo, some explanation of why this is justified would be appropriate. Another issue is that the given mortality rate gives a life expectancy of 50 years, which seems to contradict the cited source which gives the life expectancy as 59. Why this discrepancy?

If the above two issues with Table 1 turn out to have some substance then it would be good to double-check all the parameter sources.

Some minor comments:

line 90: typo

line 139-141: "supported anecdotally" is strange phrasing here, better to put this fact at the top of paragraph and use it as inspiration for the model structure

line 147: three different expose compartments not reflected in figure 2

line 190: typo

line 281: "rate of detection" is ambiguous here. Do you mean proportion of cases detected, or the rate at which new cases are reported?

line 400: I don't know what "comparable to the same calculation used" means here

line 505: typo "no" vs "not"

**Have the authors made all data and (if applicable) computational code underlying the findings in their manuscript fully available?**

Reviewer #1: Yes

Reviewer #2: Yes

PLOS authors have the option to publish the peer review history of their article (what does this mean?). If published, this will include your full peer review and any attached files.

Reviewer #1: **Yes: **SHAKIR BILAL

Reviewer #2: **Yes: **Benjamin J Singer

Figure Files:

Data Requirements:

Reproducibility:

References:

---

## [Decision Letter · Decision Letter 1]

20 Feb 2024

Dear Dr. Davis,

Thank you very much for submitting your manuscript "Comparison of stochastic and deterministic models for *gambiense* sleeping sickness at different spatial scales: A health area analysis in the DRC" for consideration at PLOS Computational Biology. As with all papers reviewed by the journal, your manuscript was reviewed by members of the editorial board and by several independent reviewers. The reviewers appreciated the attention to an important topic. Based on the reviews, we are likely to accept this manuscript for publication, providing that you modify the manuscript according to the review recommendations.

The manuscript is ready to be accepted. Please just fix the very small issue raised by one of the reviewers: "In line 211 the manuscript states that the description of the fit to Yasa Bonga data is in S1, when it is actually in S2."

Sincerely,

Alex Perkins

Academic Editor

PLOS Computational Biology

Thomas Leitner

Section Editor

PLOS Computational Biology

Reviewer's Responses to Questions

**Comments to the Authors:**

Reviewer #1: I am satisfied with authors response to my comments.

Reviewer #2: Thanks to the authors for their thoughtful responses to the issues raised in my previous review. The authors have improved the manuscript in response to my and my fellow reviewer's comments. I believe the manuscript is now ready for publication in PLOS Computational Biology.

Small note: In line 211 the manuscript states that the description of the fit to Yasa Bonga data is in S1, when it is actually in S2.

**Have the authors made all data and (if applicable) computational code underlying the findings in their manuscript fully available?**

Reviewer #1: Yes

Reviewer #2: Yes

PLOS authors have the option to publish the peer review history of their article (what does this mean?). If published, this will include your full peer review and any attached files.

Reviewer #1: **Yes: **SHAKIR BILAL

Reviewer #2: **Yes: **Benjamin J Singer

Figure Files:

Data Requirements:

Reproducibility:

References:

---

## [Editor Report · Decision Letter 2]

11 Mar 2024

Dear Dr. Davis,

We are pleased to inform you that your manuscript 'Comparison of stochastic and deterministic models for *gambiense* sleeping sickness at different spatial scales: A health area analysis in the DRC' has been provisionally accepted for publication in PLOS Computational Biology.

Best regards,

Thomas Leitner

Section Editor

PLOS Computational Biology

---

## [Editor Report · Acceptance letter]

18 Mar 2024

PCOMPBIOL-D-23-01338R2 

Comparison of stochastic and deterministic models for *gambiense* sleeping sickness at different spatial scales: A health area analysis in the DRC

Dear Dr Davis,

I am pleased to inform you that your manuscript has been formally accepted for publication in PLOS Computational Biology. Your manuscript is now with our production department and you will be notified of the publication date in due course.

With kind regards,

Anita Estes
